# In Silico Identification of Pathogenicity Effectors on *Fusarium oxysporum* f. sp. *vanillae*

**DOI:** 10.3390/biotech14030050

**Published:** 2025-06-20

**Authors:** Felipe Roberto Flores-de la Rosa, Cristian Matilde-Hernández, Nelly Abigail González-Oviedo, Humberto José Estrella-Maldonado, Liliana Eunice Saucedo-Picazo, Ricardo Santillán-Mendoza

**Affiliations:** 1Campo Experimental Ixtacuaco (CEIXTA), Centro de Investigación Regional Golfo Centro (CIRGOC), Instituto Nacional de Investigaciones Forestales, Agrícolas y Pecuarias (INIFAP), Km. 4.5 Carretera Federal Martínez de la Torre-Tlapacoyan, Tlapacoyan 93600, Veracruz, Mexico; flores.felipe@inifap.gob.mx (F.R.F.-d.l.R.); matilde.cristian@inifap.gob.mx (C.M.-H.); nabigo.888@gmail.com (N.A.G.-O.); estrella.humberto@inifap.gob.mx (H.J.E.-M.); 2Campo Experimental Edzná (CE-Edzná), Centro de Investigación Regional Sur Este (CIRSE), Instituto Nacional de Investigaciones Forestales, Agrícolas y Pecuarias (INIFAP), Km. 15.5 Carretera Campeche-Pocyaxum, Campeche 24520, Campeche, Mexico; saucedo.liliana@inifap.gob.mx

**Keywords:** proteome, phytopathogen fungi, comparative genomics, *Vanilla planifolia*

## Abstract

Vanilla is a highly valuable spice in multiple industries worldwide. However, it faces a serious problem due to a disease known as root and stem rot, caused by the fungus *Fusarium oxysporum* f. sp. *vanillae*. Little is known about the pathogenicity mechanisms this fungus employs to establish the disease, making it imperative to elucidate mechanisms such as the presence of pathogenicity effectors in its genome. The aim of the present study was to determine the presence of the *SIX* gene family in the genome of three strains of *F. oxysporum* associated with root rot: two pathogenic strains and one non-pathogenic endophyte strain. Additionally, the complete effectorome of these strains was predicted and compared to exclude effectors present in the endophytic strain. Our results show that only the *SIX9* gene is present in the strains associated with the disease, regardless of their pathogenic nature. Furthermore, no variation was observed in the *SIX9* gene among these strains, suggesting that *SIX9* is not involved in pathogenicity. Instead, we identified 339 shared effectors among the three strains, including the non-pathogenic strain, strongly suggesting that these genes are not relevant for establishing root rot but may play a role in endophytic colonization. The highly virulent strain IXF41 exhibited eight exclusive pathogenicity effectors, while the moderately virulent strain IXF50 had four. Additionally, one effector was identified as shared between these two strains but absent in the endophytic strain. These effectors and their promoters were characterized, revealing the presence of several *cis*-regulatory elements responsive to plant hormones. Overall, our findings provide novel insights into the genomic determinants of virulence in *F. oxysporum* f. sp. *vanillae*, offering a foundation for future functional studies and the development of targeted disease management strategies.

## 1. Introduction

Vanilla (*Vanilla planifolia*) is one of the most economically and culturally important crops worldwide, valued for its distinctive flavor and aroma [1]. However, its cultivation under conditions of high humidity and temperature makes it particularly susceptible to fungal pathogens, which significantly affect crop yields [2]. Among these, stem and root rot caused by *Fusarium oxysporum* f. sp. *vanillae* (*Fov*) [3] and *F. oxysporum* f. sp. *radicis-vanillae* (*Forv*) [4] stands out as the most critical disease globally, with reports from major producing countries such as Mexico [2,5], Colombia [6], Indonesia, and Madagascar [2,7].

Despite its significance, the mechanisms underlying *Fov* and *Forv* pathogenicity remain poorly understood. Unlike other *F. oxysporum* formae speciales, the damage caused by *Fov* and *Forv* appears to occur without vascular bundle colonization, likely through the release of phytotoxic compounds that induce tissue rot [2]. To date, no specific molecules—whether mycotoxins or other compounds—have been definitively associated with this process.

Among the formae speciales of *Fusarium oxysporum*, a family of pathogenicity effectors essential for disease development and associated with host specificity has been identified: the *SIX* (Secreted in Xylem) genes [8]. It has been demonstrated that this gene family confers pathogenicity to a non-pathogenic strain [9], and R proteins targeting *SIX* genes induce disease resistance [10]. However, for the *Fov*–vanilla interaction, no *SIX* genes have been identified as responsible for disease establishment. On the contrary, Solano-de la Cruz et al. [11] did not identify any homologs of these genes in the root infection transcriptome, suggesting they are not required for root rot development.

In addition to the *SIX* genes, recent research efforts have focused on identifying different pathogenicity effectors in the proteome or secretome of phytopathogenic fungi [12], which could help identify molecular targets for resistance induction [13]. Effectors are identified based on specific biochemical characteristics, such as a size of less than 400 amino acids, sequences rich in cysteine residues, the presence of a signal peptide, and the absence of transmembrane domains [14]. The collection of these effectors is known as the effectorome [15]. Identifying the effectorome of pathogenic strains provides deeper insights into the biochemical mechanisms these strains use to establish disease in different hosts [16].

In this study, we identified in silico potential pathogenic effectors from the sequenced, assembled, and annotated genomes of three *F. oxysporum* strains associated with vanilla root rot: two with differing virulence levels and one endophyte strain, reported recently by our work team [17]. By analyzing the in silico effectorome of these strains, we aim to identify potential pathogenicity-associated proteins, shedding light on the molecular mechanisms of *F. oxysporum*–vanilla interactions and contributing to the development of effective disease control strategies.

## 2. Materials and Methods

### 2.1. Genome and Proteome Database

The genome and proteome databases of the strains *F. oxysporum* f. sp. *vanillae* IXF41 (highly virulent) and IXF50 (moderately virulent), as well as the endophyte strain *F. oxysporum* IXF53, were retrieved from NCBI GenBank under Bioproject number PRJNA855480. Our research team previously reported the pathogenicity of these strains and their genome sequencing [17].

### 2.2. Synteny and SIX Gene Search in F. oxysporum Associated with Vanilla Root Rot

The linearity and synteny of the genomes of the three studied strains were determined against the genome of *F. oxysporum* f. sp. *lycopersici* 4287 [18] as a reference, obtained from NCBI (Bioproject number PRJNA640265). Mauve 2.1.0a1 software (https://darlinglab.org/mauve/mauve.html accessed on 13 November 2024) was used with default settings. Additionally, the sequences of the *SIX1–SIX14* genes from *F. oxysporum* f. sp. *lycopersici* 4287 were retrieved, and a BLASTn analysis was performed on the three genomes under study. The identified *SIX* genes were aligned, and a phylogenetic analysis was conducted against those from other formae speciales employing MEGA X software [19].

### 2.3. Identification and Comparison of Canonical Pathogenicity Effectors

Canonical effectors were identified in the proteomes of *F. oxysporum* f. sp. *vanillae* IXF41 and IXF43, as well as the endophyte strain *F. oxysporum* IXF50. The criteria used to classify a protein as a pathogenicity effector were according to [14,15,16] and included the following: (1) protein length between 10 and 400 amino acids, (2) presence of at least four cysteine residues, (3) presence of a signal peptide, and (4) absence of transmembrane domains. To identify and isolate sequences with the first two characteristics, the EffHunter algorithm [14] was employed. Once the protein sequences were isolated, the presence of a signal peptide was determined using the online tool SignalP 5.0 (https://services.healthtech.dtu.dk/services/SignalP-5.0/ accessed on 25 November 2024) with default settings. The sequences showing a signal peptide were analyzed for transmembrane domains using TMHMM 2.0 (https://services.healthtech.dtu.dk/services/TMHMM-2.0/ accessed on 27 December 2024), and those with at least one transmembrane domain were discarded. Finally, the filtered sequences were analyzed to predict whether they exhibited cytosolic or symplastic effector characteristics using EffectorP v.3.0 software (https://effectorp.csiro.au/ accessed on 6 January 2024).

The sequences identified as canonical effectors were compared among the three strains under study to identify those present only in the *F. oxysporum* f. sp. *vanillae* strains, as well as those associated with high virulence levels. Effector proteins identified in all three genomes were considered part of the core effectorome. Available annotations were used for this core, and a text analysis was performed based on the annotations. Additionally, a Venn diagram was created to compare strain-specific effectors and identify those directly associated with virulence.

### 2.4. Alignment, Phylogenetic Analysis, and Structural Prediction of Pathogenicity Effectors

The sequences exclusive to the two *F. oxysporum* f. sp. *vanillae* strains were aligned using the Muscle algorithm, and the most parsimonious tree was inferred without implicit weights, utilizing 1000 bootstrap replicates for resampling. This analysis was conducted in MEGA X software [19]. The 3D structural predictions of the protein sequences were performed using the AI-based AlphaFold3 software (https://alphafoldserver.com/ https://alphafoldserver.com/ accessed on 15 January 2025) [20].

### 2.5. Identification of Cis-Regulatory Elements in the Promoter Regions of Identified Effectors

The coding loci for the proteins associated with pathogenicity were identified, and 2000 bp upstream of the start codon were isolated from the genomes of the IXF41 and IXF43 strains, representing the promoter region. Once the promoter sequences were isolated, cis-regulatory elements were identified using the PlantCARE database [21] (https://bioinformatics.psb.ugent.be/webtools/plantcare/html/ accessed on 22 January 2025).

## 3. Results

### 3.1. Identification of SIX Genes and Synteny

The search for the 14 *SIX* genes described in *F. oxysporum* f. sp. *lycopersici* 4287 using BLASTn analysis revealed only one sequence of 758 bp corresponding to the *SIX9* gene across the three genomes (pathogenic and endophyte strains). *SIX9* was totally conserved among the *F. oxysporum* f. sp. *vanillae* and endophyte strains (Figure 1A). The phylogenetic analysis based on the coding sequences of the *SIX9* gene recovered from the three strains under study, as well as from other *F. oxysporum* formae speciales, yielded 10 equally parsimonious trees. From these, a strict consensus tree was constructed. The resulting topology showed that the *SIX9* sequences from strains IXF41, IXF50, and IXF53 grouped into a well-supported independent clade, distinct from the other formae speciales (Figure 1B). All evidence suggests that the *SIX9* gene is not associated with the pathogenic features of *Fov*.

The synteny analysis revealed that the scaffolds of the IXF41, IXF50, and IXF53 genomes are syntenic with the core chromosomes of the Fol4287 genome. However, no synteny was detected for the accessory chromosomes (Chr. 3, 6, 14, and 15) in any of the three strains. This finding is unexpected for *F. oxysporum* f. sp. *vanillae* strains (Figure 2). Therefore, these results indicate that there is no evidence of the presence of accessory chromosomes in the genome of *F. oxysporum* f. sp. *vanillae*.

### 3.2. Identification of the Core Effectorome in F. oxysporum Associated with Vanilla Roots

The in silico proteomes of the previously mentioned strains were analyzed. For the strain *Fov* IXF41, the proteome contained 18,139 sequences; for *Fov* IXF50, it contained 18,143 proteins; and for the endophyte strain *F. oxysporum* IXF53, the proteome included 18,136 protein sequences. After filtering, 350, 349, and 346 protein sequences with pathogenicity effector characteristics were identified in strains IXF41, IXF50, and IXF53, respectively.

When comparing the effectors identified across the three strains, 339 proteins were found to be shared, which were considered the core effectorome of *F. oxysporum* associated with vanilla roots (Figure 3A). Additionally, two proteins were found to be exclusively shared between strains *Fov* IXF41 and *Fo* IXF53, while five proteins were exclusively shared between strains *Fov* IXF50 and *Fo* IXF53. This suggests that these proteins are not related to pathogenicity; however, they are not considered part of the core effectorome since they are not present in all three strains. Also, specific effectors on the IXF53 genome were not observed.

Based on the available annotations for the 339 proteins, a text analysis was performed and represented in a word cloud (Figure 3B). The word cloud prominently featured concepts associated with cell wall component degradation, highlighting terms such as “lyase,” “cutinase,” “hydrolase,” and “esterase,” among others. Additionally, it emphasized carbohydrate-associated concepts like “glycosyl,” “pectate,” and “acetylxylan.” However, the word cloud also prominently included terms suggesting that the core effectorome proteins are not yet fully annotated, as evidenced by the presence of terms like “probable,” “unknown,” or “uncharacterized.”

The results of the core effectorome annotation analysis demonstrate that these effectors are associated with root cell wall degradation to facilitate colonization. However, the evidence suggests that these effectors are not associated with pathogenicity against vanilla.

### 3.3. Characterization of Putative Pathogenicity Effectors

From the effector identification analysis across the three genomes, the highly virulent strain *Fov* IXF41 was found to have eight unique specific effectors, while the *Fov* IXF50 strain contained four specific pathogenicity effectors. Both strains share one effector absent in the endophyte strain, referred to by its identifier in the *FOV* IXF41 strain (*Fov*_IXF41_031029754.1.1) (Figure 4A). These specific sequences were aligned, and a phylogenetic analysis, along with conserved domain identification, was performed. A single most parsimonious tree (L = 2121) was retrieved, showing phylogenetic homology between *Fov*_IXF41_031037941.1.1 and *Fov*_IXF50_031039538.1.1 (Figure 4A). However, no conserved domains were identified in either sequence (Figure 4B).

Some proteins in *Fov* IXF41 were grouped into clades (Figure 4A), but conserved domains were dissimilar among clade members, except for *Fov*_IXF41_031032351.1.1 and *Fov*_IXF41_031038191.1.1, which shared three out of four identified conserved domains (Figure 4B). Proteins from *Fov* IXF50 did not form specific clades (Figure 4A), and conserved domains were only identified in *Fov*_IXF50_031030885.1.1 and *Fov*_IXF41_031029754.1.1, the latter being present in both pathogenic strains of *F. oxysporum* f. sp. *vanillae* (Figure 4B).

### 3.4. Biochemical and Structural Prediction of Putative Effectors

Some biochemical characteristics of the proteins identified as pathogenicity effectors were predicted. The sequence lengths ranged from 88 to 378 amino acids, and the molecular weights (MWs) ranged from 9924.11 to 40,439.83 g mol^−1^. The predicted isoelectric points (pI) ranged from 5.29 to 9.86, with most proteins showing values below 7. Additionally, six proteins had instability index (II) values below 40, indicating high stability, while the aliphatic index (AI) values ranged from 30.63 to 98.64, showing variable thermotolerance. GRAVY values indicated that all proteins, except *Fov*_IXF50_031039538.1.1, were hydrophilic.

Moreover, prediction analyses suggested that, of the 13 sequences analyzed, 9 are cytoplasmic effectors, while only 4 are apoplastic. The results of the biochemical prediction are summarized in Table 1.

The AlphaFold3 AI platform was used to predict the three-dimensional structure of the 13 identified proteins. Figure 5 shows the structure of the eight exclusive proteins identified in the highly virulent *Fov* IXF41 strain. The structures reveal segments with high confidence in the three-dimensional prediction (blue), generally in the most folded regions. Conversely, the structure of other residues shows low confidence, corresponding to poorly structured regions (yellow-orange). The protein *Fov*_IXF41_03102935857 exhibited the highest confidence in its structural prediction, with values of II = 22.42, suggesting high stability, and AI = 69.78, suggesting high thermotolerance. It is also hydrophilic and cytoplasmic (Table 1), making it an important candidate for future evaluations.

### 3.5. Identification of Cis Elements in Promoter Region of Effectors

Up to 2000 bp were isolated upstream of the start codon of proteins identified as pathogenicity effectors, and cis-regulatory elements were identified in this region. The promoters analyzed contained between 24 cis-regulatory elements, in the protein *Fov*_IFX41_031032351.1.1, and 47 elements in *Fov*_IXF41_031039331.1.1. These regulatory elements responded to various internal and external stimuli. Notably, all promoters contained elements responsive to plant hormones (Figure 6), including auxins, methyl jasmonate (MeJA), gibberellins, abscisic acid, and salicylic acid, with MeJA response elements being particularly abundant. Additionally, light, low-temperature, and meristem elongation response elements were observed in all promoters.

## 4. Discussion

### 4.1. The Pathogenic Ability of F. oxysporum f. sp. vanillae Toward Vanilla Is Independent of the SIX Effectors

In recent years, research on the molecular mechanisms of the various formae speciales within the *F. oxysporum* species complex has focused primarily on the identification, variation, and role of *SIX* genes in their interactions with respective hosts [8]. *SIX* genes have also been utilized as molecular markers to discriminate between formae speciales [22] and for the early detection of the pathogen [23].

Our findings show that, of the 14 *SIX* genes currently reported across different formae speciales of *F. oxysporum*, only the *SIX9* gene was identified. Interestingly, this gene was present in all three strains analyzed, including the non-pathogenic endophyte strain, and no sequence variation was observed among them. Previous studies have reported that *SIX9* is highly conserved among strains of *F. oxysporum* f. sp. *vasinfectum* and is associated with increased virulence [24]. Similarly, highly virulent strains of *F. oxysporum* f. sp. *pisi* carry the *SIX9* gene, although its in planta expression is not consistently observed, suggesting that *SIX9* is not essential for pathogenicity [25].

An important aspect of *SIX9* is its chromosomal location. In other formae speciales, such as *F. oxysporum* f. sp. *cepae*, it is typically located on accessory chromosomes [26]. However, our results strongly suggest that the *F. oxysporum* strains associated with vanilla root rot, both pathogenic and endophytic, lack accessory chromosomes. Instead, the *SIX9* gene appears to be part of the core genome of *F. oxysporum*.

Phylogenetic analysis further indicates that the clade formed by *F. oxysporum* f. sp. *vanillae* strains is basal to other formae speciales. This finding suggests that the *SIX9* sequences identified in this study, which appear to play no role in pathogenicity, may be ancestral to the specialized *SIX9* sequences observed in other formae speciales. This hypothesis warrants further investigation in future studies.

The evidence presented in this work shows the absence of the remaining 13 *SIX* genes in *F. oxysporum* f. sp. *vanillae*. This aligns with the absence of accessory chromosomes found in the strain *F. oxysporum* f. sp. *lycopersici* 4287 [18], where *SIX* genes are located on chromosome 14. Similarly, *SIX* genes have been identified in accessory chromosomes in *F. oxysporum* f. sp. *cepae* [26] and *F. oxysporum* f. sp. *lactucae* [27]. Experimental evidence shows that horizontal transfer of accessory chromosomes confers virulence to endophyte strains due to the transfer of *SIX* genes, which act as pathogenicity effectors [28,29].

The absence of *SIX* genes in vanilla-pathogenic strains could be associated with the domestication level of vanilla compared to other crops [30]. Vanilla remains under very low domestication, as the use of improved varieties through traditional cultivation techniques is relatively recent [31], and genetically improved materials have not been widely disseminated to exert selection pressure on *F. oxysporum* populations [32].

Our results provide important evidence for understanding the origin of *SIX* genes and suggest that their function is associated with the domestication level of the host.

### 4.2. Identification of the Core Effectorome in F. oxysporum Associated with Vanilla Root

Numerous studies have revealed that the genome of *F. oxysporum* is composed of two distinct components: (1) the core genome, which includes all the genetic material inherited vertically and is essential for the fungal life cycle, and (2) accessory chromosomes, which, while not crucial for the fungal life cycle, provide the ability to infect specific hosts and are acquired through horizontal transfer [28].

However, no clear distinction has been made between effectors that are essential for establishing the interaction between *Fusarium* and the plant and those responsible for the development of pathogenicity. Currently, all identified proteins with the characteristics of size, presence of a signal peptide, absence of transmembrane domains, and cysteine residues are considered pathogenicity effectors [33].

The focus of this study was to use *F. oxysporum* f. sp. *vanillae* strains with varying levels of virulence toward vanilla, including a non-pathogenic endophyte strain. This approach allowed the identification of 339 proteins with effector-like characteristics shared among the three strains. These findings suggest that these proteins are not associated with virulence but rather with the ability to establish an interaction between the fungus and vanilla roots. Therefore, we propose considering these proteins as the “Core Effectorome,” representing effectors indispensable for the establishment of the fungus–plant interaction.

The analysis of annotations for the 339 proteins in the core effectorome supports the hypothesis that these effectors are necessary for establishing the interaction, as they are predominantly associated with carbohydrate degradation in the root. For example, one of the main functional annotations is “lyase” activity, which has been linked to the production of cell wall-degrading enzymes [34,35]. These enzymes can act as Microbe-Associated Molecular Patterns (MAMPs) and be detected by the plant to activate pattern-triggered immunity (PTI), as reported for the pectate lyase enzyme of *Fusarium sacchari* [36].

Additionally, other lytic enzymes, such as glycosyl hydrolases, have been reported to play a significant role in the establishment of endophytic *Fusarium* by modulating the immune response and activating PTI [37,38]. Therefore, it is crucial to further investigate the core effectorome in other formae speciales, as well as in non-pathogenic endophyte strains, to determine which effectors are indispensable for the establishment of the symbiotic interaction between *Fusarium* and the plant. Moreover, studies on the functionality of core effectors are needed to understand their role in modulating the plant immune response.

### 4.3. Identification of Putative Pathogenic Effectors in F. oxysporum f. sp. vanillae

After excluding core effectors, 13 proteins absent in the endophyte strain were identified, which are considered putative pathogenicity effectors. Of these 13 effectors, only 1 is shared between both strains. These proteins exhibited a wide range of biochemical characteristics, such as pI; however, most were found within the range of 6–9, which has been reported as optimal for fungal effectors to move into the cytosol [39].

Moreover, most of these proteins exhibit predicted features with AI values > 49.9 and II values < 40, suggesting high protein stability and hydrophilicity, characteristics typically associated with pathogenicity effectors [40]. Additionally, 3D structure prediction analyses showed that current algorithms still present many regions with low confidence levels in the modeling, which may be linked to limited experimental knowledge of these effectors.

Further studies are needed to elucidate the functionality and mechanisms of the proteins identified in this study in the establishment and progression of vanilla root rot.

### 4.4. The Promoters of Putative Pathogenic Effectors Contain Cis Elements Responsive to Plant Hormones

The analysis of cis-regulatory elements in the promoters of the putative pathogenicity effectors revealed numerous elements responsive to various stimuli. Notably, plant hormone-responsive elements were identified, including auxin-responsive elements. Several studies have reported interactions between auxins and fungal pathogenicity effectors [41,42,43], including those involving pathogens of the *Fusarium* genus [44].

Additionally, regulatory elements responsive to salicylic acid (SA) were observed. SA has been reported to induce resistance to infection by *F. oxysporum* f. sp. *vanillae* (*Fov*) over prolonged periods [45]. Abscisic acid (ABA)-responsive elements were also identified in several effector promoters. ABA has been shown to act as a virulence factor in phytopathogenic fungi, as well as a plant signaling molecule [46]. Thus, it may play a significant role in the expression of effector-encoding genes identified in this study.

On the other hand, all the promoters of the identified effectors exhibit numerous methyl jasmonate (MeJA)-responsive elements. This is particularly significant, as this phytohormone, derived from jasmonic acid, is essential during the development of infections by necrotrophic fungi [47], as reported in the *Fusarium oxysporum* f. sp. *vanillae* (*Fov*)–vanilla interaction [48].

Unfortunately, knowledge of how plant hormones influence the expression of pathogenicity effectors remains very limited. However, the present findings highlight a valuable opportunity to further investigate the molecular mechanisms underlying plant–pathogen interactions.

## 5. Conclusions

In the present study, we observed evidence that the genome of *F. oxysporum* f. sp. *vanillae* lacks accessory chromosomes based on synteny analysis. Furthermore, in line with this, we did not observe 13 of the 14 *SIX* genes identified in other formae speciales, with the exception of the *SIX9* gene. However, this gene is also found in the *Fo* IXF53 strain, which is an endophyte non-pathogenic strain, and the *SIX9* gene sequence is completely conserved across the three strains under study; the in silico analysis indicated that this gene does not participate in virulence against vanilla. Therefore, we can conclude that *SIX* genes are not necessary for *F. oxysporum* f. sp. *vanillae* to cause disease in vanilla.

We identified proteins that meet the criteria to be considered canonical pathogenicity effectors; however, we observed that 339 of these proteins are shared among the three strains, including the non-pathogenic endophyte, suggesting that these effectors do not play a role in the establishment of disease, but are instead associated with the establishment of the interaction between the fungus and the plant. This is supported by the annotation of these proteins being primarily associated with cell wall degradation. We propose the term “Core Effectorome” for these effectors that are not associated with pathogenicity and are present in non-virulent strains.

On the other hand, we identified 13 effector proteins exclusive to the pathogenic strains: 8 in the highly virulent strain, 4 in the moderately virulent strain, and 1 protein shared exclusively between these two. Their predicted characteristics align with previous reports of pathogenicity effectors. However, further studies are necessary to elucidate the role of these genes in the establishment of the root rot disease in vanilla.

Overall, our findings provide novel insights into the molecular basis of pathogenicity in *F. oxysporum* f. sp. *vanillae*, and establish a valuable genomic foundation for future studies aimed at understanding and mitigating root and stem rot in vanilla.

## Figures and Tables

**Figure 1 biotech-14-00050-f001:**
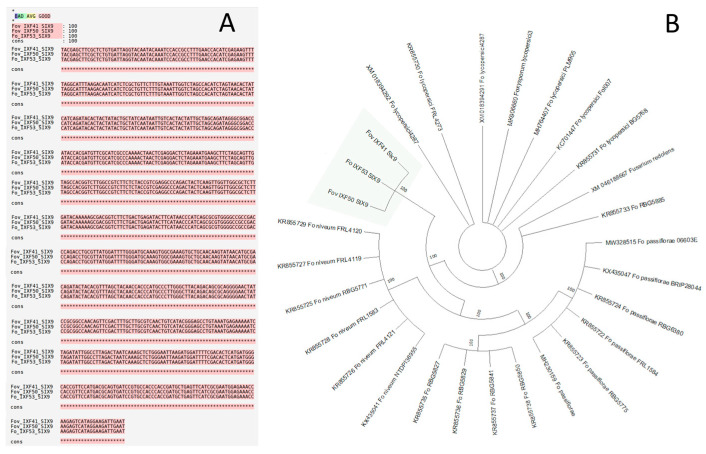
Alignment and phylogenetic tree of the *SIX9* gene from the strains of vanilla rot root. (**A**) Alignment of the *SIX9* gene sequence from the three strains under study, including two strains of *Fusarium oxysporum* f. sp. *vanillae* with different virulence levels (IXF41 and IXF50) and a non-pathogenic endophytic strain of *F. oxysporum* (IXF53) associated with vanilla roots. (**B**) Phylogenetic tree reconstructed from *SIX9* gene sequences from different formae speciales of *F. oxysporum*. The highlighted clade corresponds to the strains from this study, which form a basal clade relative to the other *F. oxysporum* formae speciales.

**Figure 2 biotech-14-00050-f002:**
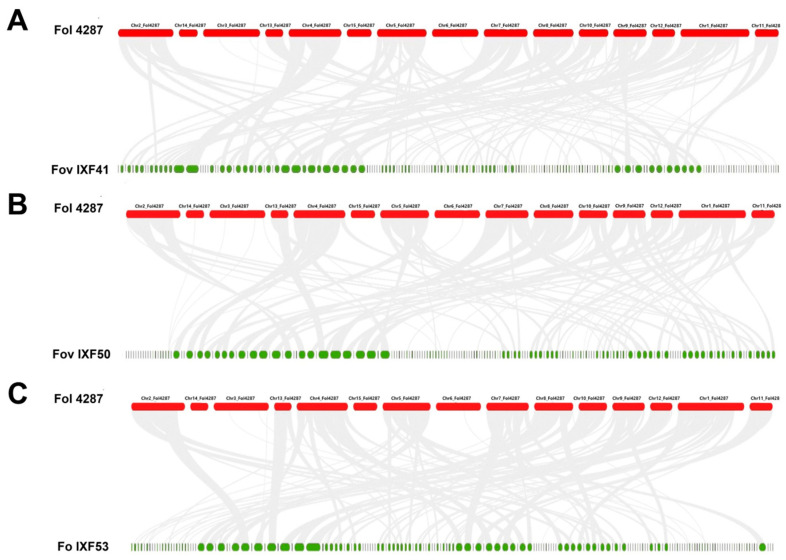
Graphical representation of synteny among the three *F. oxysporum* strains under study compared to *F. oxysporum* f. sp. *lycopersici* 4287 (Fol 4287). (**A**) Fol 4287 vs. *Fov* IXF41. (**B**) Fol 4287 vs. *Fov* IXF50. (**C**) Fol 4287 vs. *Fo* IXF53. Gray lines indicate core genome sinteny.

**Figure 3 biotech-14-00050-f003:**
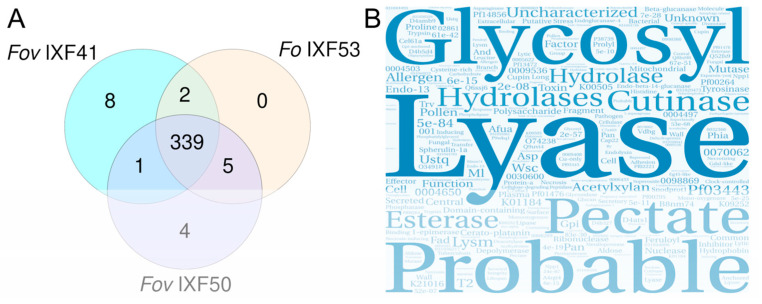
Identification of the core effectorome of *F. oxysporum* strains associated with vanilla rot roots. (**A**) Venn diagram of proteins with pathogenicity effector characteristics identified in each strain. It shows that the three strains share 339 proteins, which we consider the core effectorome. (**B**) Word cloud displaying the main categories observed within the annotation of the core effectorome.

**Figure 4 biotech-14-00050-f004:**
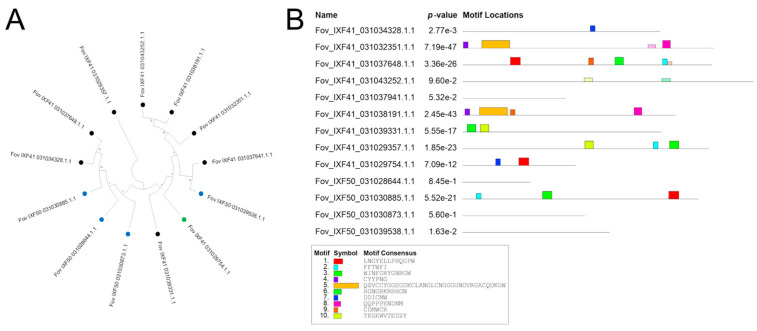
Phylogenetic and analysis of conserved motifs of pathogenicity-associated effectors. (**A**) Phylogenetic tree of the 13 pathogenicity-associated proteins identified in the *Fov* IXF41 strain (black terminal) and *Fov* IXF50 strain (blue terminal), as well as the protein shared between both strains (green terminal). (**B**) Analysis of conserved motifs in the 13 pathogenicity-associated proteins identified in the two strains of *F. oxysporum* f. sp. *vanillae*.

**Figure 5 biotech-14-00050-f005:**
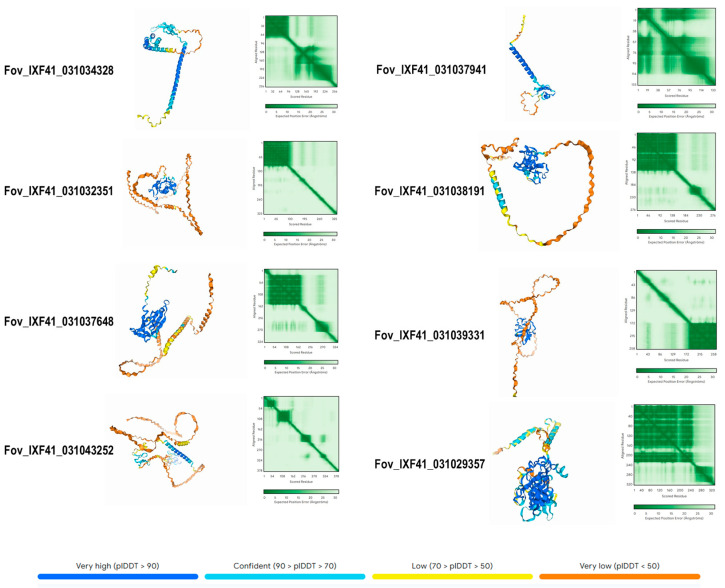
Predicted 3D structure of the eight pathogenicity-associated effector proteins exclusive to the highly virulent *Fov* IXF41 strain. The confidence value of the predicted structure is displayed according to a color-coded scheme.

**Figure 6 biotech-14-00050-f006:**
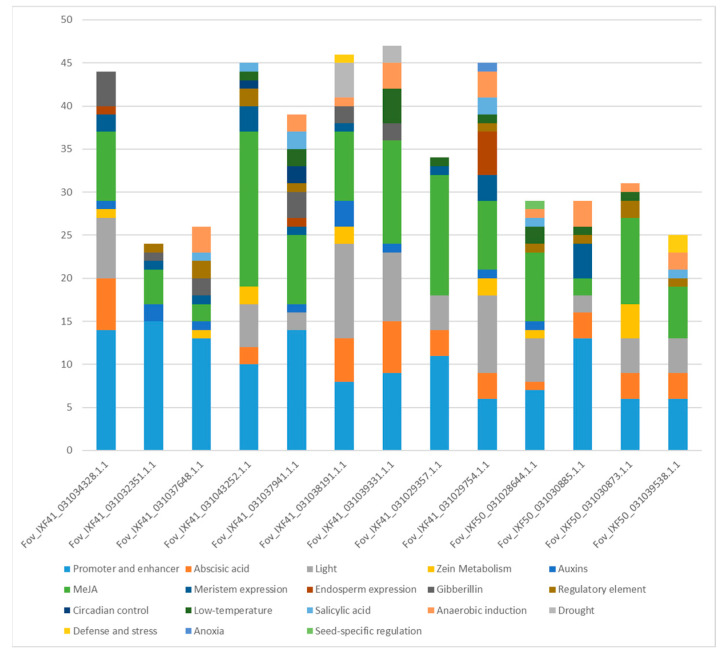
Number of cis-regulatory elements in the promoter regions of the 13 pathogenicity-associated effector proteins identified in the two strains of *F. oxysporum* f. sp. *vanillae*.

**Table 1 biotech-14-00050-t001:** Predicted biochemical characteristics of putative pathogenicity effector proteins in two strains of *F. oxysporum* f. sp. *vanilla.*

Protein	aa ^1^	MW ^2^	pI ^3^	II ^4^	AI ^5^	GRAVY ^6^	Effector Type ^7^
*Fov*_IXF41_031034328.1.1	257	27,552.25	7.94	35.50	67.90	−0.342	Cytoplasmic
*Fov*_IXF41_031032351.1.1	327	34,886.48	5.29	41.17	52.02	−0.633	Cytoplasmic
*Fov*_IXF41_031037648.1.1	324	35,352.66	5.92	25.69	88.43	−0.186	Apoplastic
*Fov*_IXF41_031043252.1.1	378	40,439.83	9.01	45.85	49.92	−0.683	Cytoplasmic
*Fov*_IXF41_031037941.1.1	134	14,782.06	9.24	43.94	63.36	−0.265	Cytoplasmic
*Fov*_IXF41_031038191.1.1	277	29,170.01	4.70	47.74	52.53	−0.482	Apoplastic
*Fov*_IXF41_031039331.1.1	258	27,552.76	5.54	50.08	54.11	−0.795	Cytoplasmic
*Fov*_IXF41_031029357.1.1	320	35,667.71	5.39	22.42	69.78	−0.707	Cytoplasmic
*Fov*_IXF41_031029754.1.1	147	16,372.57	6.31	51.77	83.40	−0.418	Cytoplasmic
*Fov*_IXF50_031028644.1.1	88	9924.11	6.44	32.21	42.05	−1.034	Cytoplasmic
*Fov*_IXF50_031030885.1.1	307	33,405.00	6.31	23.31	79.48	−0.171	Apoplastic
*Fov*_IXF50_031030873.1.1	159	18,385.14	8.77	30.39	30.63	−1.597	Cytoplasmic
*Fov*_IXF50_031039538.1.1	191	26,121.29	9.86	42.77	98.64	0.468	Apoplastic

^1^ Number of amino acids; ^2^ molecular weight in g mol^−1^; ^3^ isoelectric point; ^4^ instability index; ^5^ aliphatic index; ^6^ grand average of hydropathicity; ^7^ according to EffectorP 3.0.

## Data Availability

The original contributions presented in this study are included in the article. Further inquiries can be directed to the corresponding author(s).

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
