# Peer review of "In Silico Identification of Pathogenicity Effectors on *Fusarium oxysporum* f. sp. *vanillae"

_biotech, 2025, doi:10.3390/biotech14030050_

Round 1
Reviewer 1 Report
Comments and Suggestions for Authors
The manuscript presents a detailed in silico investigation into the effectorome of three Fusarium oxysporum strains associated with Vanilla planifolia root rot, including pathogenic and non-pathogenic variants. The study offers novel insights into the molecular mechanisms of fungal pathogenicity and endophytism, emphasizing the differential presence of effector proteins and the absence of canonical SIX genes, with a particular focus on the conserved presence of SIX9.
Overall, the topic is relevant and well-framed within the context of plant–pathogen interactions and genomic effector prediction. The manuscript is methodologically sound, employing multiple bioinformatic tools for effector identification, structural modeling, and cis-regulatory element analysis. The definition of a "Core Effectorome" shared among strains, regardless of pathogenicity, adds conceptual clarity and contributes to our understanding of the basal elements required for fungus–host compatibility, rather than virulence alone.
To strengthen the manuscript further, the authors may consider elaborating on the potential functional divergence of the shared effectors through comparative transcriptomics or expression profiling under infection conditions. Additionally, functional validation of the identified virulence-associated effectors through knockout or heterologous expression systems would significantly increase the impact and translational value of the findings.
The discussion is well-structured, integrating current literature effectively. However, given the significance attributed to hormone-responsive cis-elements, a deeper exploration of potential cross-talk between fungal effectors and plant hormonal pathways could enrich the interpretative framework. Including a schematic model summarizing effector function and promoter regulation would improve the clarity and accessibility of the findings.
In conclusion, the manuscript provides valuable data and hypotheses that merit publication following minor improvements. It represents a noteworthy contribution to the field of phytopathology and effector biology.
Author Response
Response to Reviewers
We thank the reviewers for their careful reading of our manuscript and for their insightful comments and suggestions, which have helped us to improve the quality and clarity of the work. Below, we address each of the comments raised by the reviewers and indicate the corresponding modifications made to the manuscript. All changes have been highlighted in yellow in the revised version.
- To strengthen the manuscript further, the authors may consider elaborating on the potential functional divergence of the shared effectors through comparative transcriptomics or expression profiling under infection conditions. Additionally, functional validation of the identified virulence-associated effectors through knockout or heterologous expression systems would significantly increase the impact and translational value of the findings.
Response: We appreciate this valuable suggestion. Indeed, transcriptomic and functional analyses are currently underway in our laboratory to explore the expression dynamics and biological roles of the identified effectors under infection conditions. However, these experiments are still in progress, and we anticipate that the results will form the basis of a future, separate publication. We thank the reviewer for highlighting this important direction for further investigation.
- The discussion is well-structured, integrating current literature effectively. However, given the significance attributed to hormone-responsive cis-elements, a deeper exploration of potential cross-talk between fungal effectors and plant hormonal pathways could enrich the interpretative framework. Including a schematic model summarizing effector function and promoter regulation would improve the clarity and accessibility of the findings.
Response: The identification of hormone-responsive cis-regulatory elements in the promoters of effector genes is indeed one of the most intriguing findings of our study. While there is growing evidence that fungal effectors can influence plant hormonal homeostasis, the mechanisms by which plant hormones might regulate effector gene expression during colonization remain largely unexplored. Due to this current gap in knowledge, we believe it would be premature to propose a schematic model at this stage. Nevertheless, we agree that this is a promising area for future research, and we are actively considering experimental strategies to address this interaction in upcoming studies. We thank the reviewer for this insightful suggestion.
Reviewer 2 Report
Comments and Suggestions for Authors
The authors characterized the SIX gene family and effector profiles of three Fusarium oxysporum strains associated with vanilla root rot, including two pathogenic isolates and a non-pathogenic endophytic strain. This study not only advances our understanding of vanilla disease resistance but also highlights the broader applicability of the identified multi-tiered effectors in other host-pathogen systems, underscoring their potential for translational research. The manuscript is clearly written, logically structured, and adheres to high academic standards. It is recommended for acceptance after addressing the following minor revisions.
1. Abstract & Conclusions Enhancement. To improve the impact of this work, the scientific significance of the study should be explicitly highlighted in both the abstract and conclusions.
2. Clarification on Pathogenic Effector Criteria. In lines 97–100, the authors should either justify the four criteria used for classifying proteins as pathogenic effectors or cite relevant literature supporting this classification framework.
3. Figure Quality Improvement. The resolution of Figures 1, 2, and 4 needs enhancement, as the current quality makes the text within these figures difficult to read.
4. AlphaFold 3 Reference Suggestion. While reference [20] is cited (lines 20–121), including the direct web link to AlphaFold 3 would further assist readers in accessing this resource.
Author Response
Response to Reviewers
We thank the reviewers for their careful reading of our manuscript and for their insightful comments and suggestions, which have helped us to improve the quality and clarity of the work. Below, we address each of the comments raised by the reviewers and indicate the corresponding modifications made to the manuscript. All changes have been highlighted in yellow in the revised version.
- Abstract & Conclusions Enhancement. To improve the impact of this work, the scientific significance of the study should be explicitly highlighted in both the abstract and conclusions.
Response: We greatly appreciate this insightful comment. As suggested, we have revised both the abstract and the conclusions to more explicitly highlight the scientific significance and broader impact of our findings. These changes have been incorporated into the manuscript and are highlighted in yellow for the reviewer’s convenience.
- Clarification on Pathogenic Effector Criteria. In lines 97–100, the authors should either justify the four criteria used for classifying proteins as pathogenic effectors or cite relevant literature supporting this classification framework.
Response: This comment was attended and highlighted in yellow in the text.
- Figure Quality Improvement. The resolution of Figures 1, 2, and 4 needs enhancement, as the current quality makes the text within these figures difficult to read.
Response: The figures will be submitted to the editors as original high-resolution files to ensure the highest quality in the final publication.
- AlphaFold 3 Reference Suggestion. While reference [20] is cited (lines 20–121), including the direct web link to AlphaFold 3 would further assist readers in accessing this resource.
Response: This comment was attended